# A Natural Language Processing System for National COVID-19 Surveillance in the US Department of Veterans Affairs

**Alec B Chapman**[1,2]**, Kelly S Peterson**[1,2]**, Augie Turano**[3]**, Tamára L Box**[4]**,**
**Katherine S Wallace**[5]**, Makoto Jones**[1,2]

[1] Veterans Affairs (VA) Salt Lake City Health Care System
[2] Division of Epidemiology, University of Utah
[3] VA Office of EHR Modernization
[4] VA Office of Clinical Systems Development and Evaluation (CSDE)
[5] VA Office of Biosurveillance, VA Central Office, Washington, DC

## Abstract

Timely and accurate accounting of positive cases has been an important part of the response to the COVID-19 pandemic. While most positive cases within Veterans Affairs (VA) are identified through structured laboratory results, some patients are tested or diagnosed outside VA so their clinical status is documented only in free-text narratives. We developed a Natural Language Processing pipeline for identifying positively diagnosed COVID-19 patients and deployed this system to accelerate chart review. As part of the VA national response to COVID-19, this process identified 6,360 positive cases which did not have corresponding laboratory data. These cases accounted for 36.1% of total confirmed positive cases in VA to date. With available data, performance of the system is estimated as 82.4% precision and 94.2% recall. A public-facing implementation is released as open source and available to the community.

## 1 Introduction

A robust pandemic response is contingent on timely and accurate information (Morse 2012). During the COVID-19 pandemic, public health institutions have established surveillance systems to monitor and track case counts over time.

COVID-19 is typically diagnosed using laboratory tests. The test results are frequently used as a source for surveillance systems. However, such systems typically only capture laboratory results from the same healthcare system. Patients may also be diagnosed with COVID-19 in the community, such as in external hospital networks or drive-through testing. These patients may potentially be missed by laboratory-based surveillance methods, leading to these patients not being represented in overall case counts.

Patient health information needed for biosurveillance is often recorded in free-text narratives in the Electronic Health Record (EHR) (Chapman et al. 2011), offering an alternative source of COVID-19 status when structured lab evidence is absent.

In this work we developed a Natural Language Processing (NLP) system to extract potential positive COVID-19 cases from clinical text within the Department of Veterans Affairs (VA). Following review by a clinical expert, positively identified patients are included in official VA surveillance counts. Since the VA EHR includes data from hospitals and clinics across the United States, this system enables a unique capability for collecting data for national surveillance purposes.

## 2 Background

Manual information gathering draws effort away from patient care priorities and can impede timely and effective responses to public health threats. Automated approaches for processing clinical notes have been applied for public health purposes when data is needed as quickly as possible.

Gesteland et al (2003) developed an automated syndromic surveillance system using clinical text to identify anomalies in symptoms as rapidly as possible. Several examples in the literature have utilized clinical text including chief complaints to perform early detection of infectious disease (Brillman et al. 2005; Chapman, Dowling, and Wagner 2004; Ivanov et al. 2003; Matheny et al. 2012; Pineda et al. 2015).

Typical data sources for COVID-19 surveillance include government announcements, scientific publications, and news articles (Xu et al. 2020) Most literature to date for NLP related to COVID-19 has involved public data sources such as

research publications (Wang et al. 2020). Others have examined social media sources including Twitter to examine sentiment or misinformation related to the virus (Rajput, Grover, and Rathi 2020; Singh et al. 2020). In this work, the objective was to identify the diagnosis of COVID-19 in clinical documents to report complete case counts of the disease for public health surveillance in VA.

## 3 Methods

### 3.1 Dataset

Veterans Health Administration (VHA) includes medical centers and clinics across the United States [1]. The VA Corporate Data Warehouse (CDW) includes electronic clinical data for these sites in a unified architecture. This work included clinical data in 2020 between January 1 and June 15.

### 3.2 NLP Pipeline

The primary objective of our NLP system is to classify whether a clinical document contains a positive COVID-19 case. To do this, we designed a rule-based pipeline which extracted target entities related to COVID-19, asserted certain attributes for each entity, and finally classified documents as either positive or negative based on the entities within the document. We prioritized minimizing false negatives in order to identify as many positive cases as possible. However, as the volume of data increased, it became important to reduce false positives in order to minimize manual chart review.

The pipeline was implemented in Python using the spaCy framework[2]. All processing steps except for tokenization, part-of-speech tagging, and dependency parsing were implemented using custom spaCy components, a feature available in version 2.0 and later. Each component may contain its own rules or knowledge base. Several components are available as part of medSpaCy[3], an open source project for clinical NLP using spaCy, and a publicly available version of the pipeline is released on GitHub[4].

The following describes each of the custom components in the pipeline, shown visually in Appendix A:

- **Preprocessor**: Modifies the underlying text before text processing. This step removes semi-structured templated texts and questionnaires which can cause false positives and replaces certain abbreviations and misspellings to simplify later processing steps.
- **Target Matcher:** Extracts entities related to COVID-19 based on linguistic patterns. This includes terms such as *"COVID-19"*, *"novel coronavirus"*, *"ncov"*, and *"SARS-COV-2"*.
- **Context:** Identifies semantic modifiers and attributes such as negation, uncertainty, and experiencer. This step was performed using cycontext [5], a spaCy implementation of the ConText algorithm (Chapman, Dowling, and Chu 2007). Figure 1 shows a visualization of the ConText algorithm.
- **Sectionizer:** Detects section boundaries in the text, such as *"Visit Diagnoses"* or *"Past Medical History"*.
- **Postprocessor:** Modifies or removes entities based on business logic. This component allows the pipeline to handle edge cases or more complex logic using the results of previous components.
- **Document Classifier:** Assigns a label of "Positive" or "Negative" to each document based on the entities and attributes extracted from the text.

The following is a brief description of classification logic at both entity level and document level. Entities are excluded if any of the following attributes are present:

- Uncertain
- Negated
- Experienced by someone other than the patient

Entities are marked as "positive" when any of the following conditions are met:

- Associated with a positive modifier, such as *"diagnosed with"* or *"is positive"*
- Occurring in certain sections of a note, such as *"Diagnoses:"*
- Mentioned with a specific associated condition, such as *"COVID-19 pneumonia"*

[1] https://www.va.gov/health/

[2] https://spacy.io/

[3] https://github.com/medspacy

[4] https://github.com/abchapman93/VA_COVID-19_NLP_BSV

[5] https://github.com/medspacy/cycontext

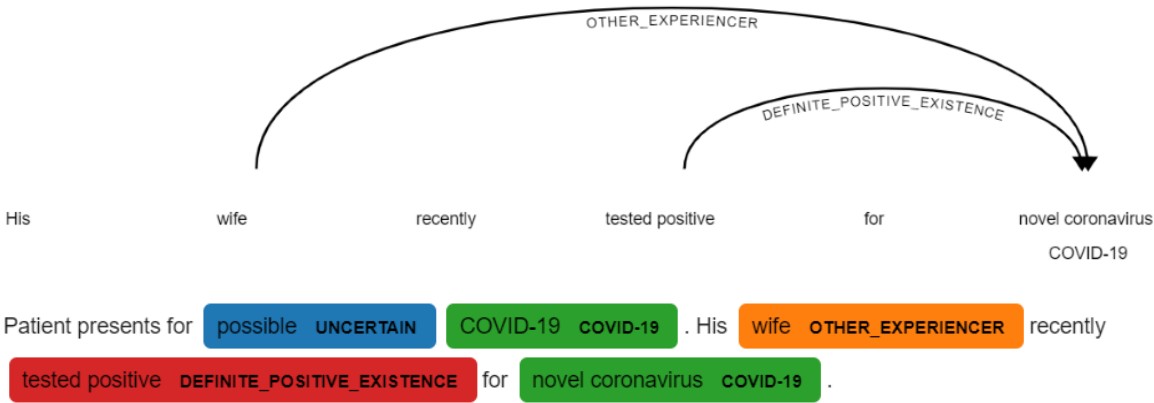

Figure 1. Visualizations provided in medSpaCy allowed us to view the output of our system and inspect linguistic patterns in the text. Target and modifier concepts are highlighted in text and arrows between them show relationships indicating whether the patient experienced COVID-19.

Based on the entities and corresponding attributes, we then classify the document as "Positive" or "Negative". In our current implementation, a document is classified as "Positive" if it has at least one positive, non-excluded entity.

### 3.3 Deployment

Our system was deployed to process clinical notes in VA CDW beginning January 21, 2020, the day after the first case was confirmed in the United States (Holshue et al. 2020). All documents containing keywords related to COVID-19 were included in document processing. Documents were retrieved and processed regularly to facilitate daily operations.

### 3.4 Clinical Review

When a patient's document was classified by text processing as positive, the document was reviewed by a clinical validator. Using an internally developed web-based tool, reviewers viewed a marked-up summary of the processed clinical documents. If the patient fit a clinical definition of COVID-19, the reviewer accepted the suggestion and the patient was added to VA's COVID-19 counts.

Due to an increasing volume of data and limited resources for review, later iterations accelerated validation and improved precision by assigning documents to "High" and "Low" priority groups using other indicators such as a relevant ICD-10 code. This allowed reviewers to prioritize review of those patients who were likely to be valid cases and to minimize the review of false positives.

## 4 Results

### 4.1 Document Processing

Keywords such as *coronavirus, novel coronavirus, COVID-19, SARS-CoV-2*, and others were found in 17 million documents in VA CDW between

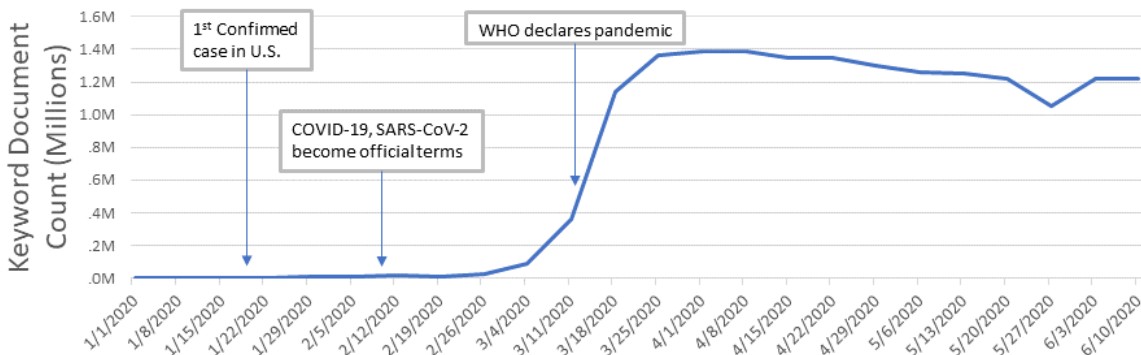

Figure 2. Frequency of documents matching COVID-19 related keywords from January through June 15, 2020. Some key dates are marked for reference.

January 1 and June 15, 2020. The median document length of this document set was 1,383 characters. Figure 2 shows the weekly volume of documents matching these keywords.

The phrase *novel coronavirus* was first observed in clinical notes the week of January 15. On February 11, 2020, World Health Organization (WHO) announced terminology of *SARS-CoV-2* for the virus and *COVID-19* as the disease it causes (World Health Organization 2020a). On March 11, WHO declared the COVID-19 situation as a pandemic (World Health Organization 2020b). In our dataset, the term COVID-19 occurred nearly 50,000 times the week of March 11 and increased to over 250,000 mentions the following week.

As of June 15, 2020, our system had processed documents from 3.6 million patients. Table 1 presents several illustrations of example text processed and classified by our system. After clinical review, a total of 6,360 patients without laboratory evidence were confirmed to be positive for COVID-19. This accounted for 36.1% of the total 17,624 positive cases identified in VA at the time.

| Text Classifications | |
|---|---|
| *Positive* | "Patient admitted to hospital for respiratory failure secondary to ***COVID-19***." |
| | "Diagnoses: ***COVID-19 B34.9***" |
| | "The patient reports that they have been diagnosed with ***COVID-19***." |
| *Negative* | "Requested that patient be screened for ***COVID-19*** via telephone." |
| | "Studies have shown that some ***COVID-19*** patients have prolonged baseline." |
| | "Has the patient been diagnosed with ***COVID-19***? Y/N" |

Table 1. Examples of positive and negative classified text.

**4.2 System Performance**

To evaluate the performance of our pipeline, we estimated precision and recall. Due to constraints, we calculated precision at a document level and recall at a patient level.

For precision, we manually reviewed 500 randomly selected documents classified as positive with an entry date on or later than May 1. We considered a document a true positive if the patient was stated to have been positive for COVID-19 and thus appropriate to review for validation.

Measuring recall is more complicated as the actual number of positive cases is not known. To estimate recall, we evaluated performance of our system for patients with positive laboratory results and at least one document containing previously mentioned keywords. We considered recall to be the percentage of these patients who had at least one document classified as positive by our system. All positive COVID-19 laboratory results completed between May 1 and June 15 were included in this analysis.

Our review yielded an estimated document-level precision of 82.4%. Estimated patient-level recall was 94.2%. Appendix B shows examples and explanations of incorrectly classified texts. One common cause of false positives was template texts such as screenings or educational information which contained phrases such as "confirmed COVID-19" but did not actually signify that the patient was positive. Several errors were referring to COVID-19 practices or the pandemic more generally, such as "COVID-19 infection control protocols". Other errors were caused by incorrectly linked targets and modifiers, resulting in marking a non-positive entity as positive or failing to mark an entity as excluded.

One source of false negatives was positive modifiers which were not linked to mentions of COVID-19. The scope for linking targets and modifiers was set to be one sentence based upon observation that linguistic modifiers typically occurred in the same sentence as a target concept. This error can be propagated by text formatting such as erroneous new lines which cause incorrect sentence splitting.

**5 Discussion**

In this work we described the development and application of a Natural Language Processing system for COVID-19 surveillance in a national healthcare system in the United States. We demonstrated that NLP combined with clinical review can be leveraged to improve surveillance for COVID-19. Within the VA surveillance system,

over one third of total known cases were identified by NLP and clinical review, with the remainder being identified through structured laboratory data. This capability validated that NLP can provide significant value to such a surveillance system, which requires a timely and sensitive case count.

Our system achieved high recall while still maintaining acceptable precision. Leveraging a rule-based system allowed defining narrow and specific criteria for what is extracted. Rules were iteratively developed to filter out irrelevant documents while still identifying positive cases.

Additionally, the flexibility of a rule-based system allowed us to add new examples and adapt to new concepts as they emerged. This was critical in the COVID-19 response, as the pandemic remains a dynamic and evolving situation. For example, the terms *COVID-19* and *SARS-CoV-2* were not announced until weeks after the surveillance system had been deployed, but requirements dictated immediate addition to our system. Similarly, changes in the clinical documentation such as new clinical concerns and semi-structured template texts required quick response and modification.

Due to the continuously changing nature of COVID-19, we required a system which permitted rapid and flexible development. While other mature clinical NLP systems exist, such as cTAKES and CLAMP (Savova et al. 2010; Soysal et al. 2018), we elected to develop this system using the features and flexibility of the spaCy framework. Rapid iteration permitted reviewing documents for errors, directly making changes to rules, and then evaluating them without compiling or reloading. Visualizations such as Figure 1 were useful to troubleshoot rule development and understand the linguistic patterns.

One limitation of this work is the evaluation of system performance. Our primary objective in this effort was to serve Veterans and provide complete public health reporting. The goal of chart review was to identify all positive patients rather than to create a reference set. Precision and recall metrics presented here are estimates using sampling and available structured data.

In future work, we plan to evaluate machine learning methods to improve identification of positive cases. A machine learning classifier could potentially improve our current system by improving document classification accuracy and identifying high-probability cases for review. This was not feasible in early stages of the response since there were very few known cases and no existing reference set. We have now identified thousands of possible cases which could be included in a training set for a supervised classifier. However, as stated previously, our clinical review did not equate to creating a reference set. Specifically, clinical reviewers did not always assign negative labels to reviewed cases which would be needed for training a supervised model. However, we believe that with additional validation and review, a machine learning classifier has the potential to augment our system's performance.

## 6 Conclusion

We have developed a text processing pipeline and utilized it to perform accelerated review of COVID-19 status in clinical documents. This approach was dynamic and allowed us to adapt to an evolving situation where vocabulary and clinical understanding continued to emerge with high data volume. Rapid implementation and iteration permitted reaction to shifting clinical documentation and evidence. This pipeline accelerated review of patient charts such that 36.1% of confirmed positive cases in a VA surveillance system were identified using this capability.

## Acknowledgments

We thank Christopher Mannozzi, Gary Roselle, Joel Roos, Joseph Francis, Julia Lewis, Richard Pham, Shantini Gamage, VA Business Intelligences Services Line (BISL), VA Office of Clinical Systems Development and Evaluation (CSDE), VHA Healthcare Operations Center (HOC), VHA Office of Analytics and Performance Integration (API), and VA Informatics and Computing Infrastructure (VINCI) Applied NLP.

We also thank the members of CSDE BASIC (Biosurveillance, Antimicrobial Stewardship, and Infection Control) for their invaluable contributions to this work.

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

## Appendix A: NLP Pipeline

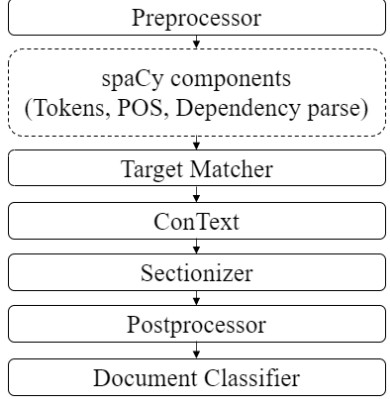

Figure 3. Diagram of components in modular text processing pipeline. Components developed in this work marked by a solid line and existing spaCy components by a dashed line.

## Appendix B: Error Analysis

| Template or educational text |
|---|
| "Do you have any:
* Fever
* **Diagnosed with** *COVID-19* in the last 14 days"

"The patient reports that they have _ _ _ _ _ **diagnosed with** *COVID-19*" |
| **Experiencer other than the patient** |
| "Veteran's *ex* tested **positive for** *COVID-19.*"

"Patient's wife is a nurse. *She* **tested positive** for *coronavirus.*" |
| **Incorrectly linked modifiers** |
| "They said he has not **presented with** any sxs of *COVID-19.*"

"Veteran with decreased **positive** *lifestyle* due to *COVID-19*." |
| **Uncertain** |
| "Admitting Diagnosis: COVID CHECK" |
| **Not relevant to patient diagnosis** |
| "TELEHEALTH SCREENING: Called to explain program **COVID-19** + Monitoring"

"**75 yo man with** telephone primary care follow-up due to *COVID-19 restrictions*." |

Table 2. Examples and explanations of false positives.

| Text formatting causes incorrect sentence splitting |
|---|
| "Employee was tested for *COVID<END OF SENTENCE>*
XX/XX/2020 and result **positive**." |
| **Positive modifier too far from target concept** |
| "Contacted Veteran for daily follow-up for *COVID-19* screening. Discussed the following: Employee tested **positive**." |
| **Incorrectly linked modifiers** |
| "**Risk for** respiratory insufficiency r/t *COVID-19*." |
| **Variations on positive modifiers not recognized by system** |
| "**62 y M** *COVID-19*" (variation of "**62 year old Male with** *COVID-19*") |

Table 3. Examples and explanations of false negatives.