# OpenReview forum: "A Natural Language Processing System for National COVID-19 Surveillance in the US Department of Veterans Affairs"
_aclweb.org/ACL/2020/Workshop/NLP-COVID — NLP-COVID-2020_

### Official Review · AnonReviewer1 · 2020-06-29
**A Natural Language Processing System for National COVID-19 Surveillance in the US Department of Veterans Affairs**

**Rating:** 6
**Confidence:** 5

**Review:**

The paper proposes a natural language processing pipeline for identifying positively diagnosed COVID19 patients, which has been deployed as part of the VA national response to COVID19. The objective of the NLP system is to simply detect if a clinical document contains a positive COVID19 case.

The paper is very USA-specific. One of the motivations of the study is the fact that community-level testing and drive-through testing are often not counted in numbers derived from healthcare systems. Thus, the authors propose an approach that detects these cases from clinical texts within the Department of VA. The motivation could be made stronger by including numbers of people whose COVID19 status are encoded in free text only. What percentage of the patients may potentially be missed by laboratory-based surveillance methods?
It is unclear if the problem of not counting certain positive cases also applies to the VA. Do VA health networks not report laboratory test based COVID19?

Preprocessor: ‘remove problematic template text’ and ‘normalize lexical variants’ are very vague descriptions of what is actually being done. Normalizing lexical variants can be as simple as simply stemming and as complex as mappings multi-word variants to standard IDs (which of course is a very hard problem).
Target matcher: Is exact matching used? What is the rate of misspellings within the clinical notes (i.e., how many misspellings per number of tokens?)
Document Classifier: what are the positive modifiers? How many of them are there? How often can it capture actual expressions of the diagnoses.
In Document Processing, how many keywords are actually used? How many times are these not spelled correctly?

What are the actual sections (incl. number of sections)?

The number of patients without laboratory evidence but with evidence only in clinical notes is quite large—and it justifies the development of such a pipeline.

The causes for the false negatives are quite straightforward and can be easily addressed. It would be good to ascertain how precision/recall would change if these simple issues (e..g, new lines) were fixed during the preprocessing phase.

It appears as though the authors did not consider inexact machine and/or semantic matching (e.g., via word2vec-type vectors). Recent research has shown that lexical similarity and semantic similarity based measures can generate data-centric variations/misspellings of noisy texts (e.g., from EHRs and social media), and such approaches have the potential to significantly improving performance—and they will not require machine learning (which would require large amounts of training data).

Since the test results for a very large number of patients already available, it would be pretty straightforward to just use those in a supervised learning framework.

---

> ### Author Response · Authors · 2020-07-06
> **Response to Reviewer #1**
>
> We thank the reviewer for reading our paper and for their thoughtful comments. Please see below for our responses to the points made in the review. We will include changes in an updated revision following any additional feedback or direction from reviewers and organizers.
>
> ---
> **The motivation could be made stronger by including numbers of people whose COVID19 status are encoded in free text only. What percentage of the patients may potentially be missed by laboratory-based surveillance methods?**
>
> **Response:** While the true percentage is not known, we report in the Results 6,360 patients who were found to be COVID-19 positive but did not have laboratory evidence and were identified using free text. This accounts for 36.1% of known positive cases in VA. We will clarify the significance of this percentage in the abstract to make it clear that these are patients without laboratory evidence.
>
> ---
> **It is unclear if the problem of not counting certain positive cases also applies to the VA. Do VA health networks not report laboratory test based COVID19?**
>
> **Response:** As described in the Introduction, laboratory test results can be captured and used for reporting when testing is performance within the reporting healthcare system. This holds in the VA as well, and per our findings ~67% of known positive cases were reported using lab results, the remainder being identified using NLP + clinical review.
>
> ---
> **Preprocessor: ‘remove problematic template text’ and ‘normalize lexical variants’ are very vague descriptions of what is actually being done.**
>
> **Response:** We will revise this text in the manuscript to offer more explanation and clarity:
> “Preprocessor: Modifies the underlying text before text processing. This step removes semi-structured templated texts and questionnaires which can cause false positives and replaces certain abbreviations or misspellings to simplify later processing steps.”
>
>  ---
> **Target matcher: Is exact matching used? What is the rate of misspellings within the clinical notes (i.e., how many misspellings per number of tokens?)**
>
> **Document Classifier: what are the positive modifiers? How many of them are there? How often can it capture actual expressions of the diagnoses.**
>
> **In Document Processing, how many keywords are actually used? How many times are these not spelled correctly?**
>
> **What are the actual sections**
>
> **Response:** Due to the operational nature of this project, we do not collect corpus statistics such as misspelling counts or phrase frequency but agree that these would be interesting research questions for future work.
> As far as the specifics of the target matching and modifying rules, we present a high-level summary of our system here due to space constraints but have made [our code available publicly](https://github.com/abchapman93/VA_COVID-19_NLP_BSV ). We hope that having access to the code can provide additional details about implementation to interested readers.
>
> ---
>
> **It would be good to ascertain how precision/recall would change if these simple issues (e..g, new lines)**
>
> **Response:** Some formatting issues, such as new lines, are difficult to fix because it is challenging to distinguish between new lines which are inserted as part of EHR formatting vs. actual new lines used to mark new sentences or sections. Preprocessing handles some cases, but applying this step too aggressively can lead to additional errors.
>
> We also include in revisions some additional examples of false negatives which illustrate more causes of errors. For example, positive modifiers which are not included in the lexicon, leading a mention to incorrectly not be marked as positive.
>
>  ---
>
> **It appears as though the authors did not consider inexact machine and/or semantic matching (e.g., via word2vec-type vectors)**
>
> **Response:** We agree that this would be a valuable step for future work in expanding the lexicon. While this was not feasible in early phases of the pandemic, where there were relatively few examples, we now have a much larger corpus which contains a great deal of variation and would benefit from this type of approach.
>
> ---
>
> **Since the test results for a very large number of patients already available, it would be pretty straightforward to just use those in a supervised learning framework.**
>
> **Response:** We agree that machine learning offers a potential to improve system performance, but we identify some challenges to implementing such a model which need to be addressed first. A revised manuscript will include text such which offers more information on our plans for machine learning and the challenges in training a classifier for our task. Due to space constraints in the OpenReview platform, please find our full response to this question in our Reply to Reviewer #3

---

### Official Review · AnonReviewer3 · 2020-06-30
**straightforward but interesting case study in building NLP system from scratch for covid-19**

**Rating:** 6
**Confidence:** 3

**Review:**

This paper describes the creation of a system from scratch for processing clinical text at the VA to detect positive Covid-19 cases. While lab tests done at the VA will be in structured data, patients who may get some care elsewhere would have positive Covid results from other care recorded only in the free text, and the authors want to be able to capture those cases.

The challenges are that there was no existing data to build training sets for machine learning, there are many mentions of Covid and related terms in contexts that do not represent positive cases (negative test results, family members, describing the reason for a televisit).

The authors of this work created an NLP pipeline built on Spacy, including Context for negation/uncertainty/experiencer, sectionizer, etc. In the end, the decision of positive/negative was made based on finding a target term that was not uncertain, negated, or experienced by someone besides the patient, and that did have a positive modifier, occur in a diagnosis section, or was used to modify another condition ("Covid-19 pneumonia").

Since this is an IR-like problem it is difficult to evaluate recall but straightforward to evaluate precision. For recall what they do is evaluate at the patient level on the subset of patients for whom they do have lab results they can use as a gold standard.

While the NLP described here is fairly straightforward rule-based methods, this is an interesting use case of bootstrapping an NLP system from nothing for a new problem where there is no retrospective data available to build ML systems. The authors tease at the end that now that they've done this bootstrapping work they have essentially a labeled dataset that could be used for an ML system. A little more focus on this aspect would've been interesting to this reviewer, even if there are no ML results to report yet.

---

> ### Author Response · Authors · 2020-07-06
> **Response to Reviewer #3**
>
> We thank the reviewer for reading our paper and for providing comments. We agree that machine learning offers a potential to improve system performance, but we identify some challenges to implementing such a model which need to be addressed first. A revised manuscript will include text such as the paragraph below which offers more information on our plans for machine learning and the challenges in training a classifier for our task:
>
> “In future work, we plan to evaluate machine learning methods to improve identification of positive cases. A machine learning classifier could potentially improve our current system by improving document classification accuracy or identifying high-probability cases for review.  This was not feasible in early stages of the response since there were very few known cases and no existing reference set. We have now identified thousands of possible cases which could be leveraged as the basis for a training set for a supervised classifier. However, as stated previously, our clinical review did not equate to creating a reference set and there are some challenges to using our dataset as a training corpus. Specifically, while many positive patients have been identified through chart review, clinical reviewers did not label negative cases, so negative labels may not be consistent. Additionally, labels are assigned at the patient level, rather than the document level, and confirmed positive patients may have a mix of both positive and negative documents. However, we believe that with additional some validation and review, a machine learning classifier has the potential to augment our system’s performance.”

---

### Official Review · AnonReviewer2 · 2020-07-07
**fairly straightforward, but apparently useful and tested on a large scale**

**Rating:** 6
**Confidence:** 4

**Review:**

This paper presents a rule-based NLP system for identifying COVID-19 cases from clinical narratives with application to 17 million documents.

Pros
- in scope for the conference
- real-life, large scale deployment
- apparent usefulness
- fairly straightforward system: the paper is accessible to a large audience

Cons
- fairly straightforward system: limited technical innovation
- ad hoc system: no/weak justification for the choice of approaches and technologies; not generalizable
- more technology than science

Overall
The significance of this work essentially comes from the practical utility of the system rather than from its originality or the innovation it brings. This work is reportable nonetheless.

Additional comments
Assuming this is the case, it would be worth emphasizing that 36.1% of confirmed positive cases in a VA surveillance system were ***solely*** identified using this capability. This is not entirely clear from the abstract or the conclusion.

---

### Decision · Program_Chairs · 2020-07-07

**Decision:**

Accept

**Comment:**

Based on the reviewer feedback, it is clear that the submission reflects a valuable real-world deployment of NLP.

It is a worthwhile contribution to the workshop. Please prepare a 10-minute presentation for Thursday July 9!

Thanks so much for your submission.

---

> ### Author Response · Authors · 2020-07-07
> **Thank you!**
>
> Thank you for the comments and acceptance! We are excited to participate in the workshop on Thursday!